# Learning Longer-term Dependencies in RNNs with Auxiliary Losses

**Trieu H. Trinh**[*]**, Andrew M. Dai, Minh-Thang Luong & Quoc V. Le**
Google Brain
{`thtrieu,adai,thangluong,qvl`}@google.com

## Abstract

We present a simple method to improve learning long-term dependencies in recurrent neural networks (RNNs) by introducing unsupervised auxiliary losses. These auxiliary losses force RNNs to either remember distant past or predict future, enabling truncated backpropagation through time (BPTT) to work on very long sequences. We experimented on sequences up to 16 000 tokens long and report faster training, more resource efficiency and better test performance than full BPTT baselines such as Long Short Term Memory (LSTM) or Transformer.

## 1 Introduction

As learning long term dependencies with recurrent networks is an important problem in machine learning, many approaches have been proposed to tackle this challenge. Well known approaches include recurrent networks with special structures (El Hihi & Bengio, 1996; Sperduti & Starita, 1997), Long Short-Term Memory (LSTM) (Hochreiter & Schmidhuber, 1997), Gated Recurrent Unit Networks (Cho et al., 2014), multiplicative units (Wu et al., 2016), specialized optimizers (Martens & Sutskever, 2011; Kingma & Ba, 2014), identity initialization and connections (Le et al., 2015), highway connections (Zilly et al., 2017), orthogonal- or unitary-constrained weights (Arjovsky et al., 2016), dilated convolutions (Salimans et al., 2017), connections (Koutnik et al., 2014), attention mechanisms (Luong et al., 2015; Bahdanau et al., 2015), skip input information at certain steps (Campos et al., 2018). As training very long recurrent networks is memory-demanding, many techniques have also been proposed to tackle this problem (Chen et al., 2016; Jaderberg et al., 2017).

Convolutional neural networks and Transformer (Vaswani et al., 2017) also mitigate or sidestep the problem of long-term dependencies, but come with other tradeoffs: they both need $O(n)$ storage during training and inference ($n$ is the input size). RNNs therefore have the special advantage when processing arbitrarily long sequences (say 1 million time steps in PTB (Marcus et al., 1994)): with a fixed BPTT length $l << n$, training and inference only require $O(l)$ and $O(1)$ storage respectively.

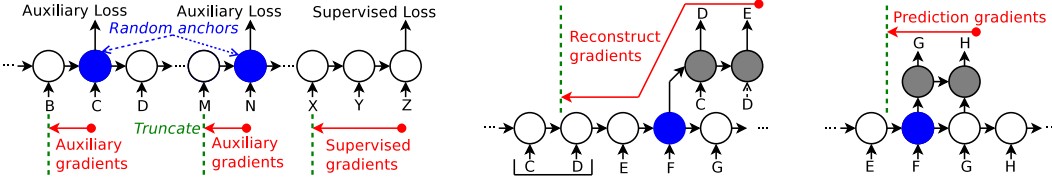

Figure 1: **Left:** We sample one or multiple anchor points along the sequence. **Middle**: We predict a random subsequence that occurs before each anchor. **Right**: We predict the next subsequence of each anchor. All gradients are truncated to a fixed length to keep BPTT cost constant.

We propose methods that are orthogonal to most aforementioned approaches, inspired by recent work in pretraining recurrent networks (Dai & Le, 2015; Ramachandran et al., 2017) to improve generalization on short sequences. We, however, focus on scalable schemes of learning long-term dependencies. An overview of our methods is shown in Figure 1. We randomly sample one or

---

[*]Work done as a member of the Google AI Residency program (`g.co/brainresidency`).

multiple anchor points, and build an unsupervised auxiliary loss at each location. In the first type of loss (*r*-LSTM), we sample a subsequence before the anchor, and ask a decoder to *reconstruct* it from the first token. In the second type (*p*-LSTM), the decoder performs future *prediction* over a subsequence starting from the anchor. If there are enough anchors, good embeddings are learnt such that supervised BPTT only needs a few steps to refine them for better classification. We pretrain our models by minimizing auxiliary losses, and then perform semi-supervised training where the sum of supervised and auxiliary losses is minimized.

## 2 EXPERIMENTS AND ANALYSIS

We consider a wide variety of datasets with sequences of varying lengths from 784 to 16384 on pixel-by-pixel image classification, using MNIST, permuted MNIST, CIFAR10 and StanfordDogs dataset[1] (Khosla et al., 2011). All images in StanfordDogs are scaled down to 8 different sizes from $40\times40$ to $128\times128$ before being flattened into sequences of pixels. We also explore a real language benchmark where the DBpedia character level classification task is chosen[2], with averaged sequence length of 300. In most experiments, anchors are sampled at frequency 1 per sequence, all hyper-parameters are fixed regardless of sequence length to demonstrate the scalability of our models. With the final goal of processing over 16000 long sequences, both supervised and auxiliary BPTT are truncated to only 300 time steps, while anchored subsequences are sampled with length 600.

An overview of Table 1 shows that our proposed auxiliary losses produce gradually larger improvement moving from MNIST to pMNIST and CIFAR10. On MNIST, *r*-LSTM and *p*-LSTM perform on par with fully trained RNNs like uRNN and LSTM. On pMNIST where more complex dependency is presented, our models easily outperform all baselines. On CIFAR10 this discrepancy is even larger. With full supervised BPTT, we obtained the best accuracy across all three datasets.

Table 1: Test accuracy (%) on MNIST, pMNIST, and CIFAR10.

|  | MNIST | pMNIST | CIFAR10 |
|---|---|---|---|
| uRNN Arjovsky et al. (2016) | 95.1 | 91.4 | N/A |
| LSTM Full BPTT | 98.3 | 89.4 | 58.8 |
| *r*-LSTM Truncate300 | 96.4 | 92.8 | 65.9 |
| *p*-LSTM Truncate300 | 95.4 | 92.5 | 64.7 |
| **r-LSTM Full BP** | **98.4** | **95.2** | **72.2** |

As training on StanfordDogs is expensive, we restrict each training session to the same resource (a single Tesla P100 GPU) and report infeasible when one training example cannot fit into memory. All models are evaluated after 100 epochs of training, with 20 epochs of pretraining for our models. We also evaluate Transformer[3] and *r*-LSTM with 20 anchors (*r*-LSTM 20x30[4]) on this benchmark. Result in Figure 2-left shows that both *r*-LSTM and *p*-LSTM exhibit the strongest resistance to the growing difficulty when sequences get longer. Although Transformer starts with best accuracy, its performance degrades fastest, while *r*-LSTM 20x30 stays at the top throughout. Gradient truncation also allows our methods to train faster with less memory usage. Specifically, Transformer and LSTM become infeasible after the 9 000 and 12 000 mark respectively, while Figure 2-right reports significant faster training for our methods against the fully trained LSTM.

On character-level DBpedia where the average sequence length is much shorter, we truncate BPTT length to only 100 and still obtain superior results[5] than comparable baselines as reported in Table 2.

In our models, BPTT cost will become relatively negligible if sequences get indefinitely long. We simulate this scenario by gradually shrinking supervised BPTT truncation down to only one and zero

---

[1]We follow the procedure suggested in Sermanet et al. (2014) to obtain a larger training set.

[2]DBpedia texts is normalized following the procedure suggested in Zhang et al. (2015).

[3]We use Tensor2Tensor (https://github.com/tensorflow/tensor2tensor) with an off-the-shelf configuration that has a comparable number of parameters to our RNNs (0.5M weights). A simple setting is adopted where output vectors are average-pooled and fed into two fully-connected layers to make predictions.

[4]To keep the overall cost of BPTT constant, we shrink the subsequences from length 600 down to 30.

[5]Following Dai & Le (2015), we turn off joint training on this dataset to obtain better result.

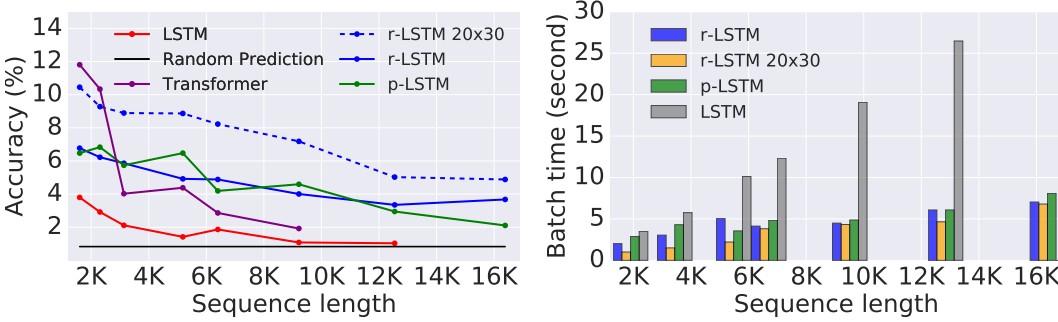

Figure 2: **Left**: Test accuracy on StanfordDogs resized to 8 levels of sequence length. **Right**: Time to train on a single mini batch.

Table 2: Test error rate (%) on character-level DBpedia document classification.

|  | Test error |
| --- | --- |
| LM-LSTM Truncate100 (Dai & Le, 2015) | 4.45 |
| SA-LSTM Truncate100 (Dai & Le, 2015) | 4.89 |
| *r*-LSTM 20x15 Truncate100 | 3.84 |
| *p*-LSTM Truncate100 | **2.85** |

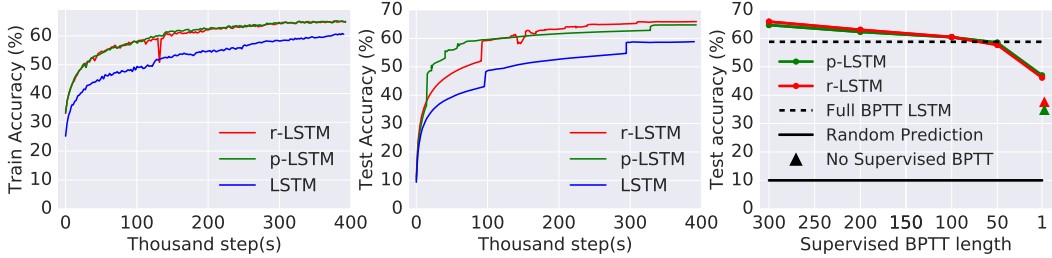

Figure 3: **Left and Middle**: Training and testing accuracy during training on CIFAR10. **Right**: Effects of shrinking supervised BPTT length on test accuracy.

step, while keeping input length fixed using the CIFAR10 dataset. Results in Figure 3-right show that our methods can still perform approximately as good as fully trained LSTM at only 50 BPTT steps. At extreme cases of one and zero-step BPTT, both *r*-LSTM and *p*-LSTM perform commendably well.

Lastly, we explain the significant gap made on almost all benchmarks between our models and a fully-trained LSTM, in terms of optimization and regularization. Figure 3-left and middle show that *r*-LSTM and *p*-LSTM training curves trace each other almost identically, while *r*-LSTM gives better result on testing data. This implies that *r*-LSTM regularizes better than *p*-LSTM. Comparing to LSTM, our methods start off with much higher training accuracy while having the same testing accuracy, revealing the optimization advantage provided by unsupervised pretraining. Later on, this optimization gap quickly becomes smaller than the corresponding testing accuracy gap. This highlights the strong regularization effect induced by optimizing the semi-supervised loss.

## 3 CONCLUSION

We propose adding unsupervised auxiliary loss to RNNs and show that this improves generalization by regularizing the model. By combining auxiliary losses with truncated BPTT, we also demonstrate that our methods can train faster than other recurrent models, while using significantly less memory. We anticipate this technique to be widely applicable to systems that process unusually long sequences.

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
