# OpenReview forum: "Learning Longer-term Dependencies in RNNs with Auxiliary Losses"
_ICLR.cc/2018/Workshop — Accept_

### Official Review · AnonReviewer4 · 2018-03-08
**The paper is hard to read and seems to be of limited value compared to Twin Networks which is easy to understand and seems easier to implement, train and tune. A comparison with their results is necessary, benchmark LM results would greatly strengthen the paper.**

**Rating:** 5
**Confidence:** 4

**Review:**

The paper suggests to train an RNN where each state can both reconstruct its future and past inputs. In the case of sequence classification, this two reconstruction objectives acts as regularizer. In the case of language modeling (LM), the first reconstruction objective is the task (?) and the past reconstruction acts as a regularizer.

I feel the contribution is limited to the reconstruction of the past since regularizing LSTM sequence classifier with a side LM task (or LM pre-training) has been done before. In the case of regularization by reconstruction of the past, I feel it would be necessary to cite and compare with Twin Networks: Matching the Future for Sequence Generation (Dmitriy Serdyuk et al, August 17) which introduce a stronger regularizer in the same spirit. There, the LSTM states have to match those of a backward language model. I would also suggest to report experiments on strong LM benchmarks such as PTB, 1billion word, wiki103.

On presentation, I feel that the paper is extremely unclear. There is no equation or even a name given for the auxiliary losses. I assumed that you maximized the likelihood of the future/previous symbols given the state, this should have to be guessed by the reader. Also, it is not clear why only a subset of points are anchors, as opposed to having auxiliary losses for all time steps.

Overall, I feel that the paper is hard to read and seems to be of limited value compared to Twin Networks which is easy to understand and seems easier to implement, train and tune. A comparison with their results is necessary, benchmark LM results would greatly strengthen the paper.

---

### Official Review · AnonReviewer2 · 2018-03-09
**An interesting idea yielding good results, but the presentation could be imporoved**

**Rating:** 8
**Confidence:** 4

**Review:**

The authors present a method to train recurrent neural networks for sequence classification tasks without having to back-propagate the loss signal through the whole sequence. To that end, they introduce two auxiliary objectives which encourage the model to keep track of important information: the hidden state at each time step is required to hold enough information to predict the next few time steps (p task) or to recall previously read symbols (r task). The authors show that if a model is trained on these tasks and on the task of interest with very limited back-propagation through time, it can match or improve on the performance of BPTT at a significantly reduced cost. The authors also compare and give some insights into the regularization properties of both auxiliary objectives.

The main issue of this paper is clarity: the entirety of the method is described in the third paragraph and Figure 1, and the lack of any kind of implementation details makes understanding what is going on a little difficult (for example, I assume that both "prediction" and "reconstruction" use a log-likelihood loss, but cannot find actual confirmation in the text). Even given the limited length of a workshop submission, adding a sentence or two there could be most helpful. Still, this is certainly a helpful method, which may hopefully be developed to work with larger label sets.

---

### Official Review · AnonReviewer1 · 2018-03-11

**Rating:** 6
**Confidence:** 3

**Review:**

In this paper, the authors propose a new method to learn recurrent networks with long-term dependencies. One difficulty of
training such models is that performing back-propagation is memory expensive (as it cannot be truncated). A few techniques
have been proposed to tackle this issue, such as the synthetic gradients (Jaderberg et al., 2016). This paper propose a new
alternative, based on auxiliary losses which are added for each sub-sequence of the BPTT algorithm. The authors consider two
unsupervised losses: a reconstruction loss (predicting previously seen tokens) and a prediction loss (predicting future tokens).
They show experimentally that these auxiliary losses make it possible to train RNNs with long dependencies on classical tasks
such a permuted MNIST or CIFAR10.

Overall, I think this paper introduces an interesting method, which is simple and efficient (based on results from the paper).
The experimental results from the papers seems strong (although I don't know the SOTA in that area). My main concerns with this paper are the following:
- First, I believe that it is a bit hard to understand the method, and more technical details would be welcomed (e.g. where are
the auxiliary losses added? at the beginning or end of BPTT sub-sequences?)
- I think it would also be interesting to have (more) experimental comparisons with previous work, such as synthetic gradients.

Pros/Cons:
+ simple yet efficient method
+ strong experimental results
- paper is hard to follow
- a bit incremental compared to pre-training (e.g. with "LM" loss)

---

### Public Comment · ~Oriol_Vinyals1 · 2018-02-17
**Please Fix Length**

Your paper violates by a few lines the 3 page limit (see https://iclr.cc/Conferences/2018/CallForWorkshops). Please send us a fixed version of your PDF at iclr2018.programchairs@gmail.com by the end of Monday, February 19th, or else we will reject your paper.

Thanks,
ICLR2018 Program Chairs

---

> ### Public Comment · ~Trieu_Hoang_Trinh1 · 2018-02-17
> **Fixed version submitted**
>
> Thank you for your request, we submitted the fixed length version to iclr2018.programchairs@gmail.com today (February 17th, 2018).

---

### Public Comment · (anonymous) · 2018-02-18
**Is the unsupervised loss applicable to Transformer as well?**

Transformer's cache cost can be upper-bounded if we modify the self-attention. It was discussed in the original paper as "restricted self-attention," and recently OpenAI announced that this kind of modification to be an important open problem, not to mention that there's no guarantee that it will work. So, I'm curious to know an answer to my question in the title.

---

### Decision · Program_Chairs · 2018-03-20
**ICLR 2018 Workshop Acceptance Decision**

**Decision:**

Accept

**Comment:**

Congratulations, your paper was accepted to the ICLR workshop.